# Wake Up! Resuscitation of Viable but Nonculturable Bacteria: Mechanism and Potential Application

**DOI:** 10.3390/foods12010082

**Published:** 2022-12-23

**Authors:** Hanxu Pan, Qing Ren

**Affiliations:** School of Light Industry, Beijing Technology and Business University, Beijing 100048, China

**Keywords:** resuscitation, viable but nonculturable state, functional bacteria, application

## Abstract

The viable but nonculturable (VBNC) state is a survival strategy for bacteria when encountered with unfavorable conditions. Under favorable environments such as nutrient supplementation, external stress elimination, or supplementation with resuscitation-promoting substances, bacteria will recover from the VBNC state, which is termed “resuscitation”. The resuscitation phenomenon is necessary for proof of VBNC existence, which has been confirmed in different ways to exclude the possibility of culturable-cell regrowth. The resuscitation of VBNC cells has been widely studied for the purpose of risk control of recovered pathogenic or spoilage bacteria. From another aspect, the resuscitation of functional bacteria can also be considered a promising field to explore. To support this point, the resuscitation mechanisms were comprehensively reviewed, which could provide the theoretical foundations for the application of resuscitated VBNC cells. In addition, the proposed applications, as well as the prospects for further applications of resuscitated VBNC bacteria in the food industry are discussed in this review.

## 1. Introduction

Since it was first discovered in *Escherichia coli* and *Vibrio cholerae* in 1982, the viable but nonculturable (VBNC) state has been a widely known phenomenon adopted by microorganisms when confronted with stressful environments [1]. Under this state, VBNC cells lose the ability to grow on routine culture medium, but their metabolic activities and gene expression capacities are retained, and the cytoplasmic membranes remain integrated [2]. However, this state is still controversial due to the suspicion that VBNC cells are actually “dying” cells with residuary metabolic activities that cannot be cultured on media, or that VBNC cells are “dead” cells with minor injuries on the membrane [3]. There are distinct differentiations between death and the VBNC state: bacterial death is the point where the injury extent is beyond the ability of a cell to resume growth; however, the VBNC state postulates a specific program of a long-term survival state rather than a short-term survival state followed by a further dead state [4]. In this circumstance, resuscitation is a keystone of the VBNC state. The resuscitation phenomenon from the VBNC state was first recognized in *Salmonella enteritidis* and *E. coli* in 1984 [5]. While providing VBNC bacteria with favorable conditions, the transition from a VBNC state to a culturable state is termed “resuscitation” [2]. Entry into the VBNC state can be considered as a survival strategy under stresses only when the VBNC cells possesses the ability to resuscitate. In other words, resuscitation is a requisite to prove the existence of the VBNC state [4].

VBNC bacteria are widely distributed in environments such as water, air, soil, foods, medical facilities, food processing procedures, and so on [1]. A large proportion of bacteria that can enter the VBNC state are pathogenic bacteria, which may still express toxins under the VBNC state or regain infectivity and pathogenicity after resuscitation, causing human illness or food spoilage [2,6]. The resuscitation of VBNC cells in foods may take place during shelf-life storage, which could be associated with foodborne outbreaks [7]. For a long time, researchers spent a lot of effort studying the risks of VBNC-state bacteria and resuscitated cells to human health. Makino et al. proved that VBNC *E. coli* O157:H7 in salted salmon roe could regain pathogenicity in germfree mice by resuscitating in the mouse intestine [8]; VBNC *E. coli* O157:H7 and VBNC *Legionella pneumophila* cells retained the ability to produce toxin genes or virulence proteins [9,10]; VBNC *Campylobacter jejuni* retained the ability to invade Caco-2 human intestinal epithelial cells in vitro [11]. Therefore, VBNC-state bacteria and, more importantly, their resuscitation, are becoming hot areas in food safety research. However, every coin has two sides. When functional bacteria enter the VBNC state, things could be different; the prevention and control of resuscitation sometimes may be switched to promotion and reasonable application, which may play key roles in ecological processes and/ or have great value in the food industry [12]. No matter for what reason, the exploration and recognition of resuscitation mechanisms is necessary and of great significance.

In this review, the different aspects of resuscitation including the confirmation strategies, resuscitation factors, resuscitation mechanisms, as well as the prospects for potential application in the food industry have been comprehensively reviewed. This review aims to provide updated and in-depth references for researchers and lay a foundation for investigations into the isolation and application of resuscitated VBNC bacteria.

## 2. Confirmation of Resuscitation from the VBNC State

Since the VBNC state is an ecologically significant state for bacteria, the ability of cells to undergo a resuscitation process from this dormant state to an actively metabolizing state must be possible to prove the existence of the VBNC state. There was skepticism that the regrowth phenomenon might be due to the very few culturable cells rather than the resuscitation of VBNC cells [13,14,15]. For example, Bogosian et al. thought that the resuscitation of VBNC-state *V. vulnificus* induced by a low temperature was the regrowth of H_2_O_2_ sensitive culturable cells [13]. To contradict the above suspicions, several strategies were adopted to exclude the impact of possibly existent culturable cells. For instance, the induced VBNC-state bacterial suspensions were diluted serially to minimize the possible existence of culturable cells before resuscitation [16,17]. When mixtures of culturable and nonculturable cells are diluted to the point where only nonculturable cells are present, the revived cells are resuscitated cells from the VBNC state [18]. In addition, antibiotics such as ampicillin were added to the medium after VBNC induction to inhibit the proliferation of remaining culturable cells, the actively growing cells during the resuscitation procedure were therefore confirmed to be resuscitated cells from the VBNC state [19]. Furthermore, the possibility of the regrowth of H_2_O_2_-sensitive culturable cells was excluded by the addition of an H_2_O_2_ scavenger including sodium pyruvates and catalases to the resuscitation medium [20,21]. Based on the above strategies, the resuscitation process from the VBNC state was confirmed and the strategies were further applied in resuscitation-related investigations. With the evidence proposed above, resuscitation is now widely accepted as the recovery of VBNC cells, which is usually determined through plate counting or turbidity measurement [6,22].

The ability to resuscitate is dependent on the persistent period of the VBNC state and external stress intensity. It was proposed that resuscitation ability was gradually impaired with a prolonged VBNC-state duration time [20,23]. After an overlong time, bacteria might even lose the ability to resuscitate [24]. Therefore, this period was defined as the “resuscitation window” [25]. In addition, Zhao et al. discovered that the resuscitation ability of VBNC state *E. coli* O157:H7 reduced significantly with an increased intensity of induction conditions [26].

## 3. Resuscitation: The Reverse Process of the VBNC State?

Plenty of factors are contributory to the resuscitation of VBNC cells, including, but not limited to, external stress removal, supplementation with peroxidases, coculturing with or being inoculated to the host of VBNC cells, and supplementation with resuscitation promoting factors (Rpfs) (Table 1). In many cases, the simple reversal of VBNC-inducing factors was sufficient to allow resuscitation, so the resuscitation process sometimes might be simply regarded as a reverse process of the VBNC state. However, it may be inexact because the removal of stressful environments sometimes may not be contributory to resuscitation [27]. In addition, other resuscitation factors such as Rpfs and autoinducers (AIs) have also implied the existence of signaling pathways to stimulate resuscitation. Therefore, the resuscitation process may be a complicated physiological process rather than the simple reverse of the VBNC sate.

VBNC cells have distinct characteristics such as declined metabolic activity, decreased or loss of pathogenicity, dwarfing, or abnormal morphology [25,28,29]. Stimulated by a variety of environmental, biological, or chemical stimuli, VBNC cells may resuscitate and recover their cell division ability with an elevated metabolic level, as well as pathogenicity and cell morphology (Figure 1). The recovery of the abnormal morphology of VBNC cells to some extent is a re-shape process, and the restored cell division ability during resuscitation from the VBNC state requires the re-synthesis of cytoplasmic proteins and cell wall peptidoglycan. Through supplementing chloramphenicol and penicillin, which inhibits protein and peptidoglycan synthesis, respectively, to the resuscitation medium, VBNC state *V. vulnificus* was found unable to resuscitate [24,30]. In addition, after the inhibition of the penicillin-binding proteins PBP1 and PBP5, which were involved in the late assembly of peptidoglycan, VBNC state *E. Faecalis* cells could not resuscitate [31]. Therefore, resuscitation is not simply a reverse process of the VBNC state; newly synthesized proteins and possibly a remodeling of the cell wall to shape a normal morphology may be necessary in this process.

**Table 1 foods-12-00082-t001:** Conditions that facilitate the resuscitation process of VBNC cells.

Resuscitation Factors	Bacterial Species	Resuscitation Conditions		References
		VBNC-State Induction Condition	Corresponding Resuscitation Condition	
External stress removal	*Arcobacter butzleri*, *Aeromonas hydrophila*, *Staphylococcus aureus*, *Vibrio vulnificus*, *E. coli*	Low temperature	Temperature up-shift	[24,32,33,34,35,36]
	*Salmonella bovismorbificans*, *Enterococcus faecalis*, *Citrobacter* sp., *V. cholerae*, *Listeria monocytogenesisolates*, *Enterococci* sp., *Pasteurella piscicida*, *Yersinia pestis*, *V. shiloi*, *V. tasmaniensis*, *V. parahaemolyticus*	Starvation	Addition of nutrients	[29,37,38,39,40,41,42,43,44,45]
	*Enterobacter cloacae*	Desiccation	Rewetting	[46]
	*E. coli* O157:H7, *S. enterica* serovar Typhimurium, *L. monocytogenes*	Low pH	Adjustment to the optimal pH	[47]
	Acetic acid bacteria, lactic acid bacteria	O_2_ deprivation	Addition of O_2_	[48]
	*E. coli* O104:H4, *Acidovorax citrulli*, *Erwinia amylovorain*	Copper	Addition of chelating agent	[49,50,51]
	*S. enterica*, *E. coli* O157:H7	Food processing techniques	Stress removal	[26,52]
Supplementation with peroxidases	Yeasts, *Ralstonia solanacearum*, *E. coli* O157:H7, *Enterococcus* sp., *Salmonella* sp. *S. aureus*, *V. cincinnatiensis*	Catalase, sodium pyruvate, SOD, GST, CAT, acetaldehyde		[53,54,55,56,57,58,59,60,61]
Host of VBNC cells	*Legionella pneumophila*, *E. coli* O157:H7, *Campylobacter jejuni*, *Helicobacter pylori*, *L. monocytogenes*, *V. cholerae* O1, *Francisella tularensis*, *E. faecalis*, *Campylobacter* sp.	Yolk sacs of embryonated eggs/1-week-old chicks, Caco-2 human intestinal epithelial cells, passage in the mouse intestine, co-culture with eukaryotic cells, injected intraperitoneally into mice, mice stomachs, co-culture with *Acanthamoeba/Castellanii/Acanthamoeba polyphaga*, ingestion by *C. elegans*, inoculated in iron-dextran-treated mice		[9,11,62,63,64,65,66,67,68,69,70,71,72,73,74,75,76,77,78,79]
Supplementation with substances that could promote resuscitation	*Salmonella typhimurium*, *E. coli* O157:H7, *Vibrio* sp., *V. parahaemolyticus*	Supplementation with autoinducer (AI)		[6,80,81,82]
	*H. pylori*, *Mycobacterium tuberculosisare*, *Rhodococcus* sp., actinobacteria, *M. smegmatis*, Sphingomonas and Pseudomonas, *Rhodococcus biphenylivorans* strain TG9^T^	Supplementation with resuscitation promoting factor (Rpf)		[83,84,85,86,87,88,89,90,91,92,93]

## 4. Mechanisms of Resuscitation

Previously, most VBNC-related studies focused on the exploration of formation mechanism [22,94,95,96], while studies on the mechanisms of resuscitation were rare. For the purpose of preventing and controlling the hidden risk caused by VBNC cells (resuscitated VBNC cells) or VBNC bacterial strain application after resuscitation, an explanation of the resuscitation mechanism was necessary. Summarized from the existing studies, such mechanisms can be classified into the following aspects: resuscitation promoting factors (Rpfs), quorum sensing, pyruvates sensing and application, and mechanisms based on global metabolism analysis.

### 4.1. Rpfs

The discovery and application of Rpf is a notable landmark in the resuscitation of VBNC cells. Rpf protein was first discovered in *M. luteus* as a bacterial cytokine, which promotes the resuscitation and growth of non-growing or dormant cells [97,98]. Rpf is a muralytic enzyme revealed by its cell wall peptidoglycan lysis ability, which contains a 70-residue domain at the C-terminal that adopts a lysozyme-like fold, and the invariant catalytic glutamate residue is conserved [99]. Similar proteins are widely occurred among other high G+C gram-positive bacteria, including corynebacterial, mycobacteria, streptomycetes, and fermicutes (contain Rpf analogues) [100]. It was reported that Rpf protein with a picomolar concentration could increase the viable cell number of dormant *M. luteus* at least 100-fold [98].

The resuscitation effect of Rpf was significant; however, its functioning mechanisms were not thoroughly studied. Through analyzing the products from mycobacterial peptidoglycan hydrolysis reactions, RpfB was found to form a complex with a protein named as resuscitation-promoting factor interacting protein (RipA) [101]. In this complex, RpfB cleaves the β-1,4-glycosidic bond between N-acetylmuramic acid (MurNAc) and GlcNAc, whereas RipA is predicted to be an endopeptidase that cleaves the stem peptide (D-iGlu-*meso*-diaminopimelic acid (Dap)) [101,102]. Both proteins colocalize at the septum of dividing cells and work synergistically to hydrolyze mycobacterial PG [103]. The complex of RpfB–RipA was reported to be inhibited by penicillin binding protein 1 (PBP1): RipA would form a complex with PBP1 and form a thick layer of PG at the septum. With the increased concentration of RpfB, RipA might exchange PBP1 for RpfB to form a new complex with a high efficiency of PG hydrolysis [104]. Some researchers thought that such a type of cell wall hydrolysis would directly stimulate VBNC cell resuscitation, since the peptide moieties of PG were crosslinked heavily in the VBNC state to resist external stresses [28,101,105]. Therefore, the recruitment of Rpf and RipA during PG remodeling is essential for cell division and resuscitation (Figure 2A). Apart from that, the PG fragments derived from cell wall hydrolysis could directly activate resuscitation [106]. However, how exactly PG fragments activate the resuscitation process remains unclear and researchers have proposed hypotheses to try to explain it. Panutdaporn et al. found that the addition of rabbit anti-Rpf Ab inhibited the resuscitation effect by Rpf, thereby suggesting that Rpf might be a signal molecule that could bind to the receptor to trigger the resuscitation process [107]. Moreover, the extracytoplasmic domain of Ser/Thr kinase PknB in *Mycobacterium tuberculosis* could bind exogenous PG fragments hydrolyzed by Rpf with its muralytic activity, which was conducive for PknB to localize at the mid-cell to stimulate growth [108] (Figure 2B). Although possible mechanisms have been proposed, more evidence is still needed to prove the activation process of PG fragments in resuscitation, which is a problem to be solved in future studies.

Other Rpf analogues were also reported to possess resuscitation-promoting abilities. The YeaZ protein in *V. parahaemolyticus*, *V. harveyi*, *S. typhimurium*, and *E. coli* has been shown to have promoting effects on VBNC-state recovery [107,109,110,111]. The *yeaZ* gene was found to be ubiquitous in the genome of bacteria such as *Salmonella* sp. and *E. coli*, which was necessary for bacterial growth [112]. Zhao et al. proposed that YeaZ exhibited protease activity, and muralytic activity was lower. Single amino acid mutation greatly affected protease activity, as well as resuscitation-promoting ability [113]. However, the impact of mutation was much less on the muralytic activity of YeaZ, and the resuscitation-promoting effect was not affected [113]. Hence, in contrast to Rpf, the promoting effect of YeaZ may be correlated with its protease activities, but its function mechanism lacks further investigation.

### 4.2. Quorum Sensing

Quorum sensing (QS) is a widespread communication system in bacteria, which is a type of population density-dependent cell–cell signaling that triggers changes in behavior when the bacterial population reaches a critical density [114]. QS signaling can result in global changes in gene expression [115]. Typically, signal molecules are continually generated with a low bacterial concentration, and the signal accumulates to a threshold concentration as the population density increases. Afterwards, the signal will interact with its receptor protein to cause a coordinated change in bacterial gene expression [115]. Such hormone-like molecules are termed as autoinducers (AI), of which there are several types, including acyl-homoserine lactone (AHL)-type signals (usually generated in G^-^ bacteria), short oligopeptide signals (in G^+^ bacteria), *Streptomyces* γ-butryolactones, and the AI-2 family (in *V. harveyi* and *S. typhimurium*) [114].

QS signaling in bacteria can orchestrate an adaption to stressful conditions, and it has been reported to play a role in the resuscitation of VBNC cells. Ayrapetyan et al. discovered that the bacterial cell-free supernatants of *V. vulnificus* containing AI-2 molecules could awake VBNC *Vibrio* populations within oysters and seawater, which was inhibited by the QS inhibitor cinnamaldehyde [A]. Previous studies have indicated that the QS system was involved in the activation of superoxide or catalase to regulate the antioxidation activities in Pseudomonas aeruginosa [116]. Furthermore, Liao et al. (2019) also suggested that the QS system triggered the expression of catalase to restore the growth of VBNC-state *S. typhimurium* [80]. In accordance with that, AI-2 was found to be useless in the resuscitation of the *rpoS* mutant of *V. vulnificus*, whose production of catalase was suppressed [82]. Hence, it was suggested that RpoS is also an important factor in AI-2-mediated resuscitation [82,117]. Based on the above results, a model was proposed (Figure 3): during resuscitation, the gradually generated AI-2 molecules synthesized by LuxS specifically bind to the periplasmic binding protein of LuxP, which forms a two-component sensing kinase system with LuxQ [118]. With a low level of AI-2, LuxQ acts as a kinase, but it acts as a phosphatase while AI-2 is at a high level. Therefore, the phosphorelay of LuxO derepresses the expression of LuxR (a transcription factor in the QS regulon), which can stimulate *rpoS* expression and subsequently induces the expression of catalase (KatG) [82]. Through this regulation, cells are allowed to persist under the toxic properties of H_2_O_2_ and revive to a culturable state [82]. To sum up, QS signaling may be critical for the resuscitation process of VBNC bacteria.

### 4.3. Pyruvates Sensing and Application

Sodium pyruvate (SP), a well-known intermediate key metabolite in glycolysis, is known to be functional in the resuscitation of VBNC cells. VBNC cells are able to grow on standard media, but they can revive on media supplemented with SP [26]. SP has long been regarded as an H_2_O_2_-degrading compound that could facilitate the resuscitation of VBNC cells under prolonged stress or the effects of toxic chemicals, such as H_2_O_2_ produced in a culture media during autoclaving [54,119,120,121]. It was suggested that VBNC cells could be resuscitated to a culturable state by SP or other substances such as catalase and superoxide dismutase, due to their H_2_O_2_- or reactive oxygen-degrading effect [20,21].

More opinions have emerged recently. Apart from being an H_2_O_2_-degrading compound, pyruvate is also a kind of carbon source that can be utilized by bacterial cells. Morishige et al. found that pyruvate and its analogue α-ketobutyrate both showed restoration activities; however, other well-known antioxidant or radical-scavenging reagents such as *N*-acetyl-L-cysteine, α-lipoate, and D-mannitol were ineffective in resuscitating VBNC *Salmonella* Enteritidis cells induced by H_2_O_2_ [57]. Through further investigation, it was implied that α-keto acids and pyruvate were incorporated by VBNC cells, which were related to the restoration of the biosynthesis of macromolecules, especially DNA, not just degrading intracellular peroxide [57]. It was later shown that pyruvate was avidly taken up by starved and cold-stressed VBNC *E. coli* cells through the high-affinity pyruvate/H^+^ symporter BstT/YhjX, which was regulated by two pyruvate-sensing hidtidine kinase response regulator systems, BtsS/BtsR and YpdA/YpdB, respectively [122]. BtsSR and YpdAB are two-component systems (TCSs) which respond to extracellular pyruvate, composed of a membrane-integrated histidine kinase (BtsS/YpdA) that can perceive pyruvate, and a cytoplasmic response regulator (BtsR/YpdB) mediates *btsT* expression [123,124]. With the import of pyruvate, cells then initiate DNA and protein biosynthesis for growth restoration (Figure 4). Therefore, VBNC cells may utilize pyruvate as an alternative carbon source and correspondingly fine-tune their transport capacities and metabolism for resuscitation.

### 4.4. Mechanisms Based on Global Metabolism Analysis

On most occasions, the reported studies on resuscitation mechanisms were based on the role of specific proteins or pathways, which may result in a less systematic and comprehensive investigation. With the extensive application of high-throughput sequencing technologies in biomolecular frontiers, more research based on omics analysis has emerged in the investigation of not only the VBNC-formation mechanism, but also the resuscitation mechanism.

Up to now, most omics studies on resuscitation from the VBNC state were conducted based on proteomics analysis. A thorough iTRAQ-based proteomic profile analysis of VBNC and resuscitating cells of the plant-pathogenic bacterium *Acidovorax citrulli* was reported, indicating that protein expression varied in the different resuscitation processes [125]. In the early stage, the proteins associated with carbon metabolism, degradation of naphthalene and aromatic compounds, and superoxide dismutase or catalase were significantly enriched, while the proteins involved in oxidative phosphorylation, bacterial chemotaxis, ABC transporting, and quorum sensing were significantly enriched at the late resuscitation stages [125]. From this point, it is evident that as the resuscitation progress proceeds, the metabolic activities may change to meet their different needs. In the early stage, heavily stressed bacterial cells try their best to cope with the adverse environments to guarantee their survival and gradually increase their metabolic activity for multiplication. With the increase in the cell number, cell-to-cell signaling is enhanced to better adapt the environment for further revival. The proteomic profile of the resuscitating *V. parahaemolyticus* was compared with the VBNC state and the exponential phase cells, revealing that the metabolic activity of resuscitated cells shared minor differences with exponential phase cells, but when compared with VBNC cells, the differently expressed cells were comprehensively upregulated, which mainly involved protein synthesis, secretion system, trans-membrane transport, adhesion, movement, and other vital processes [126]. Debnath et al. suggested that the most variably expressed proteins of resuscitating *V. cholerae* showed a combination mode of adaptive and survival responses under conditions of nutrient limitation [127]. For example, the expression of PhoX, PstB, and Xds might help in the utilization of extracellular DNA to promote growth; the expression of AhpC addressed the significance of the oxidative stress response; the upregulation of EctC, an enzyme related to the biosynthesis of ectoine that is crucial for osmoadaptation, might be a response to the long-term stress of high salinity [127]. The analysis of global metabolism provides an overall perspective of the resuscitation mechanisms, which can also be a basic foundation for further investigation of specific mechanisms.

## 5. Potential Application of the Resuscitation of VBNC Cells

Unculturable microorganisms exist as “dark matter” in ecological environments, which greatly affects the exploration and utilization of microbial resources. This unculturability is largely derived from the adverse natural habitat conditions of bacteria, since they are always inconstant, and their inherent characteristics such as the oligotrophy of water, desiccation of soil, and existence of pollutants greatly limit the normal growth of bacteria, many of which are functional species. Therefore, resuscitating VBNC bacteria could provide huge candidates for obtaining high-value strains.

At present, the resuscitation of VBNC bacteria has been proved to be effective in shaping bacteria populations. With the resuscitation-promoting ability of the Rpf protein, the abundance of specific taxa was significantly increased and 51 potentially novel bacterial species were isolated from a nutrient-rich compost soil [128]. Su et al. also reported that after resuscitation by Rpf, bacterial diversity was increased, especially in terms of functional bacterial communities [91]. Since then, more studies have emerged on resuscitating VBNC bacteria to search functional bacteria populations in samples of soil or water. Wang et al. obtained a richer species diversity while resuscitating cells with Rpf protein, and two rare actinobacteria were resuscitated and isolated [129]. It was proposed that resuscitating VBNC bacteria through adding Rpf into polychlorinated biphenyl-contaminated soil accelerated the biodegradation of Aroclor 1242, which was mainly due to the resuscitation of key bio-degraders of the *Sphingomonas* and *Pseudomonas* genera [89]. In addition, bacterial populations were shaped, and 13 strains were resuscitated and isolated from river sediments under the function of Rpf, which possessed nitrogen removal capacities [92]. Therefore, through resuscitating VBNC cells, environmental-friendly strains that possess pollution control capacities were separated for further use, whose effect was notable. The resuscitating effect was achieved through the functioning of the Rpf protein. However, the Rpf protein is mainly derived from the culture supernatants of *Micrococcus luteus* or the heterologous expression of the *Micrococcus luteus* gene [83,91,130]. Other species may also express Rpf protein and have a significant resuscitation effect, and this area is therefore worth further exploration.

The resuscitation and separation of VBNC cells in foods seems meaningful in another aspect as well. For example, in fermentative foods, the lowered pH, lack of oxygen, and particular metabolites may pose negative impacts on the normal growth of bacteria, and some of them may enter the VBNC state. The resuscitation of the VBNC-state functional strains such as flavor-producing strains, fermentation strains, and probiotics, through adding resuscitation-promoting stimuli, is of great significance, and can be a research aspect in the future (Figure 5). Therefore, inspired by the Rpf-induced resuscitation, the application of VBNC-cell resuscitation may be conducted from two aspects. Except for with Rpf, more resuscitation-promoting conditions can be used in the resuscitation of bacteria in the VBNC state; in real food or other samples, bacterial strains can be separated and identified to search strains with the *rpf* gene (or Rpf protein), so that the strains could be adopted to increase microbial diversity in the samples, through which the source of the Rpf protein can be enriched.

## 6. Conclusions

The formation of VBNC-state pathogenic bacteria is a great threat to food safety and public health. However, when it comes to functional bacteria, entering a VBNC state makes them become a hidden resource for potential industrial application. In this article, aspects of the resuscitation of VBNC cells, including the definition and confirmation of resuscitation, promoting factors, and the mechanisms of resuscitation, are thoroughly reviewed, which could lay a firm theoretical foundation for the isolation and application of VBNC-state functional populations, as well as the prevention of risks arising from VBNC-state pathogenic bacteria. Attempts to resuscitate VBNC bacteria and the specific roles of the resuscitated cells have been studied in various environments in recent years, the effects of which have been proved to be profound. However, such studies in the food area are rare. Regarding the universality of the emergence of the VBNC phenomenon in the food industry, waking up the dormant and functional population may provide a new approach to obtaining valuable microbial resources, which may have great value for the food industry. However, care should be taken regarding the corresponding concerns. The microbial communities in food products, especially in fermented foods, are always complicated, and whether the resuscitation process restores some “unfavorable” microorganisms at the same time is unknown. Therefore, we think that the resuscitation of VBNC cells may be beneficial for the isolation of rare species or functional populations from foods, but the direct supplementation of resuscitation-promoting substances into foods should be rigorously evaluated to avoid the occurrence of food safety events and serious alterations to foods.

## Figures and Tables

**Figure 1 foods-12-00082-f001:**
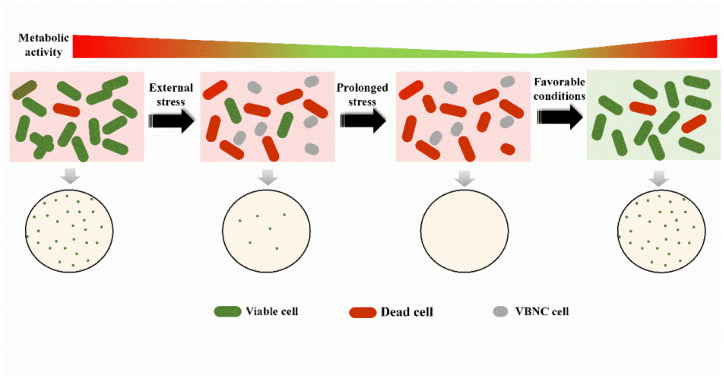
The formation of the VBNC state and its resuscitation. While confronted with prolonged stressful environments, a small proportion of viable bacteria will enter the VBNC state, under which bacteria cannot develop into colonies on their culture medium, but cellular metabolic activities are retained although significantly decreased. When provided with favorable conditions, VBNC cells will resuscitate to the viable state with recovered metabolic activity and culturability.

**Figure 2 foods-12-00082-f002:**
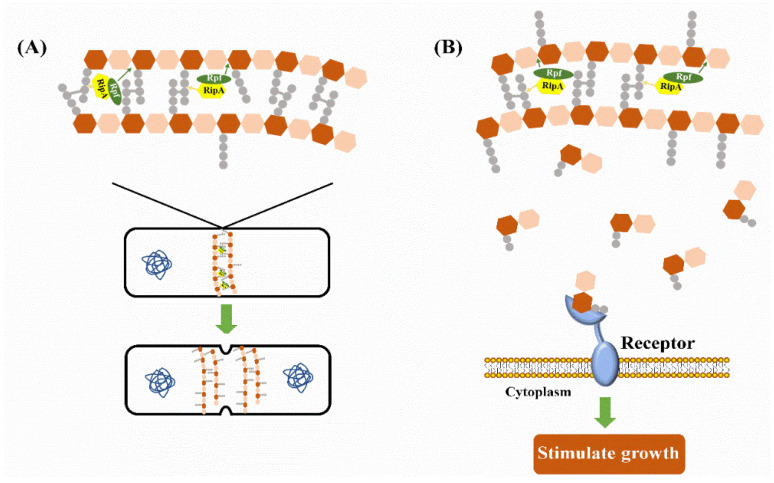
Function mechanisms of Rpf protein. (**A**) Rpf and RipA work synergistically to remodel the cell wall to promote cell division and resuscitation. (**B**) Cell wall fragments digested by Rpf and RipA serve as signaling molecules binding with a receptor to trigger resuscitation related pathways.

**Figure 3 foods-12-00082-f003:**
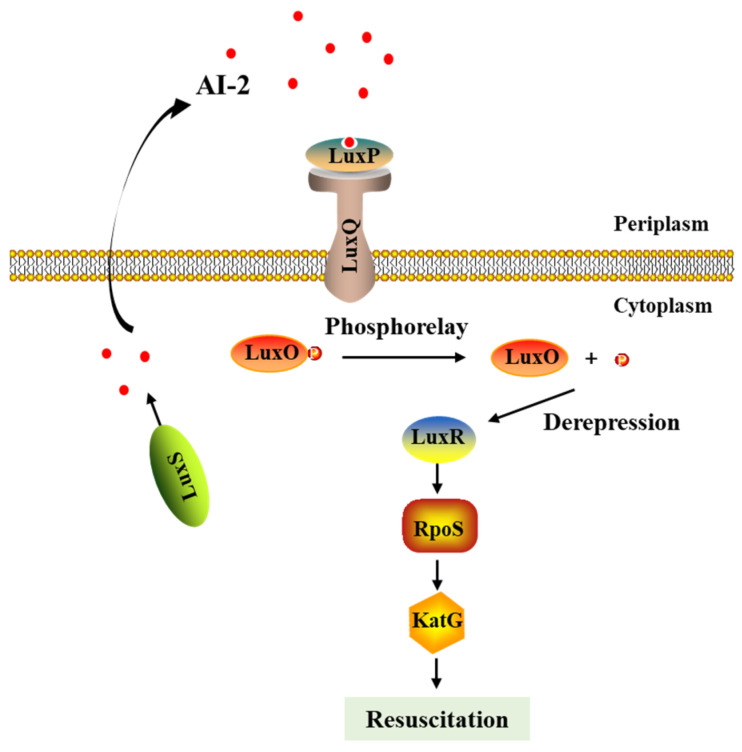
Generalized schematic of the effect of the QS phosphorelay system on the resuscitation of VBNC bacteria. The thin arrows mean promotion effects, and the thick arrow means transportation direction of AI-2.

**Figure 4 foods-12-00082-f004:**
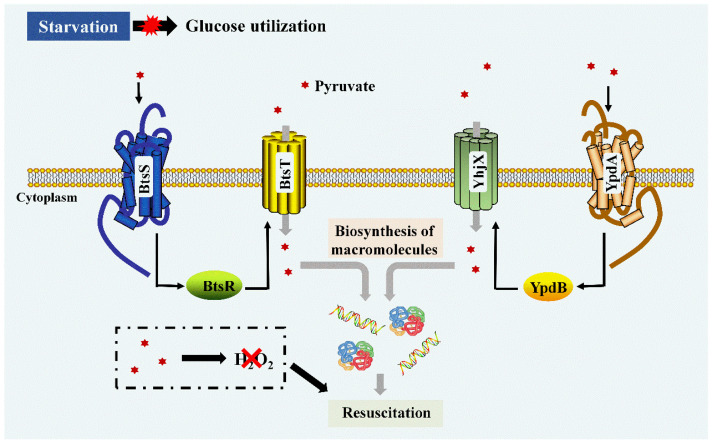
The uptake and utilization of pyruvate to resuscitate VBNC bacteria. Pyruvate was taken up through the high-affinity pyruvate/H+ symporter BstT/YhjX, which was regulated by two pyruvate-sensing hidtidine kinase response regulator systems, BtsS/BtsR and YpdA/YpdB, respectively. BtsSR and YpdAB are two-component systems which respond to extracellular pyruvate, composed of a membrane-integrated histidine kinase (BtsS/YpdA) that can perceive pyruvate, and a cytoplasmic response regulator (BtsR/YpdB) mediates *btsT* expression. With the import of pyruvate, cells then initiate DNA and protein biosynthesis for growth restoration. The black thin arrows indicate that the proteins/substances promote the synthesize of the transporter of BtsT/YhjX. The grey thick arrows mean the promotion effect of pyruvate to resuscitation through biosynthesis of macromolecules. The black thick arrows mean promotion effect of pyruvate to resuscitation through removing H_2_O_2_.

**Figure 5 foods-12-00082-f005:**
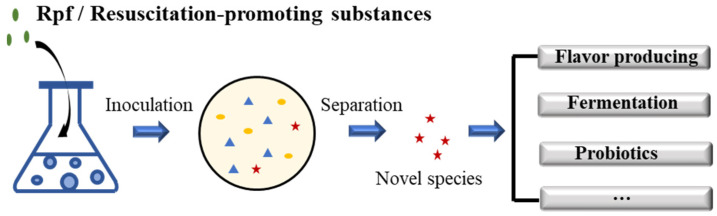
Assumption of VBNC bacteria resuscitation to separate novel species for application in the food industry.

## Data Availability

This review did not report any data.

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
