# Peer review of "Wake Up! Resuscitation of Viable but Nonculturable Bacteria: Mechanism and Potential Application"

_foods, 2022, doi:10.3390/foods12010082_

Round 1

Reviewer 1 Report

The manuscript entitled, “Wake up! Resuscitation of Viable but Nonculturable Bacteria: Mechanism and Potential Application”, is a review about different aspects of bacterial resuscitation in order to provide the updated knowledge on the isolation and application of resuscitated bacteria, also in the food industry.

The work is interesting and worthy of attention; however, it appears to be lacking in relation to the hygienic-sanitary aspects and the potential impact of VBNC bacteria on public health.

These aspects should also be explored in the concluding discussion.

Regarding the use of a VBNC fermentative flora, it is not clear how it can be controlled in a food.

In fact, it is known that the uncontrolled multiplication of some microorganisms can determine the appearance of more serious alterations of foods than any favourable effects.

Line 357, it would be better to include conclusions in the work.

Line 358,  Other literature on the subject should be cited, for instance Zhang, XH., Ahmad, W., Zhu, XY. et al. Viable but nonculturable bacteria and their resuscitation: implications for cultivating uncultured marine microorganisms. Mar Life Sci Technol 3, 189–203 (2021). https://doi.org/10.1007/s42995-020-00041-3

Reviewer 2 Report

The paper is a review, but it is well written and clear.

I have two issues to be reported,

1) only a certain number of species were touched in the paper and some not (e.g. Listeria monocytogenes, Campylobacter spp), then could be good to highlight the interest of the author in some other species, giving to the rider the complete approach to the problem.

2) the paper is not addressed specifically to food, then I kindly ask the editor to take that into consideration.

Some minor fine spelling issues should be fixed.

Round 2

Reviewer 1 Report

The changes and additions made to the manuscript are considered sufficient